# The Biosynthesis of the Monoterpene Tricyclene in *E. coli* through the Appropriate Truncation of Plant Transit Peptides

Meijia Zhao [1,2,†], Shaoheng Bao [2,†], Jiajia Liu [2], Fuli Wang [2], Ge Yao [2], Penggang Han [2], Xiukun Wan [2], Chang Chen [2], Hui Jiang [2], Xinghua Zhang [1,*] and Wenchao Zhu [2,*]

1   School of Chemical and Environmental Engineering, Shanghai Institute of Technology, Shanghai 201418, China; 15136336287@163.com
2   State Key Laboratory of NBC Protection for Civilian, Beijing 102205, China; jingwang_surgeon@yeah.net (S.B.); jiajialiu0802@163.com (J.L.); wangfuli5728@163.com (F.W.); bzyaoge@163.com (G.Y.); hanpeng1021@163.com (P.H.); xiukunwan@126.com (X.W.); chenchang15@mails.ucas.ac.cn (C.C.); ylplkmc@163.com (H.J.)
*   Correspondence: xhzhang@sit.edu.cn (X.Z.); 15811043200@139.com (W.Z.)
†   These authors contributed equally to this work.

**Abstract:** Tricyclene, a tricyclic monoterpene naturally occurring in plant essential oils, holds potential for the development of medicinal and fuel applications. In this study, we successfully synthesized tricyclene in *E. coli* by introducing the heterologous mevalonate (MVA) pathway along with *Abies grandis* geranyl diphosphate synthase (GPPS) and *Nicotiana sylvestris* tricyclene synthase (TS) XP_009791411. Initially, the shake-flask fermentation at 30 °C yielded a tricyclene titer of 0.060 mg/L. By increasing the copy number of the TS-coding gene, we achieved a titer of 0.103 mg/L. To further enhance tricyclene production, optimal truncation in the N-terminal region of TS XP_009791411 resulted in an impressive highest titer of 47.671 mg/L, approximately a 794.5-fold improvement compared to its wild-type counterpart. To the best of our knowledge, this is the highest titer of the heterologous synthesis of tricyclene in *E. coli*. The SDS-PAGE analysis revealed that lowering induction temperature and truncating the random coil N-terminal region effectively improved TS solubility, which was closely associated with tricyclene production levels. Furthermore, by truncating other TSs, the titers of tricyclene were improved to different degrees.

**Keywords:** tricyclene; tricyclene synthase; biosynthesis; transit peptide truncations; protein solubility





## 1. Introduction

Monoterpenes are a class of C10 terpenoid compounds and are widely used in the pharmaceutical, cosmetic, and fuel industries [1,2]. Tricyclene, a tricyclic monoterpene, is a valuable product that naturally occurs in the essential oils of a number of plant species, such as *Salvia aegyptiaca* [3], *Polyalthia suberosa* [4], and *Elettaria cardamomum* [5]. It was reported that plant essential oils rich in tricyclene have antioxidant [5], antitumor [4], and antimicrobial [5–7] properties. Meanwhile, it has been reported that the essential oil derived from *Agathis robusta* bark, which is abundant in tricyclene, exhibits potential efficacy against COVID-19. Utilizing Molecular Operating Environment software for in silico docking analysis, we discovered that the constituent tricyclene within the essential oil demonstrates favorable binding affinities to the active sites of the spike receptor-binding domain, main protease, and RNA-dependent RNA polymerase targets. Consequently, this component holds promise as a potent therapeutic agent for combating COVID-19 [8]. In addition, the widespread use of traditional fossil fuels as non-renewable energy has led to the intensification of the greenhouse effect and climate crisis [9]. Therefore, the development of high-energy-density biofuels is imperative for achieving a sustainable future [10]. Monoterpenes, such as α-pinene [11,12], camphene [2], and limonene [13,14], possess high energy density due to their cyclic structures, rendering them suitable as

raw materials for renewable fuels with high energy density. Consequently, the tricyclic monoterpene tricyclene may also hold potential as a precursor for advanced biofuels.

Traditionally, monoterpenes have been derived from plant biomass; however, this conventional extraction method is laborious and inefficient, necessitating the significant consumption of natural resources [15]. Therefore, the achievement of large-scale commercial production has not yet been realized. Similarly, the chemical synthesis strategy for monoterpenes was considered as a means to enhance productivity; however, it is accompanied by drawbacks such as intricate reactions, exorbitant costs, and significant environmental pollution [16,17]. Recently, the heterologous production of monoterpenes by microbial hosts has garnered significant attention due to advancements in metabolic engineering and synthetic biology [18].

The advantages of *Escherichia coli* include rapid growth, elucidated metabolic pathways, facile gene manipulation techniques, and well-established fermentation procedures [16,19]. Recently, the use of *E. coli* as a cell factory for the gram-scale synthesis of monoterpenes has been reported, such as geraniol (5.52 g/L) [20], $\alpha$-pinene(1.04 g/L) [10], and limonene (3.6 g/L) [21]. Therefore, constructing and engineering *E. coli* as a cell factory for the efficient production of tricyclene is operational. Mcdaniel et al. expressed heterologous terpene synthase in *E.coli* to produce monoterpenes, where the production level of tricyclene was at least 5% of the total monoterpene production, and indicated that tricyclene has very good octane properties and oxidative stability, so it may be more suitable as a gasoline component than other monoterpenes [22].

Isopentenyl pyrophosphate (IPP) and dimethylallyl pyrophosphate (DMAPP) are universal precursors for terpenoids [23]. The biosynthesis pathways of IPP and DMAPP include the methylerythritol-4-phosphate (MEP) and the mevalonate (MVA) pathways. The MEP pathway is generally found in bacteria such as *E. coli*, while the MVA pathway is generally found in eukaryotes such as *Saccharomyces cerevisiae*. Martin et al. successfully constructed the heterologous MVA pathway in *E. coli* for the first time and obtained a high-yield strain of amorphadiene, which can be used as a platform host for the production of general terpenoids [24]. Afterward, due to the regulatory mechanisms of the native host limiting the expression of the MEP pathway, the heterologous expression of the MVA pathway in *E. coli* to synthesize monoterpenes, such as $\alpha$- Pinene [25], limonene [26], sabinene [27], and $\gamma$-terpinene [28], was reported. Similar to other monoterpenes, GPPS catalyzed the conversion of IPP and DMAPP into geranyl diphosphate (GPP), which subsequently underwent transformation by TS to yield tricyclene.

Plant terpene synthases may possess an N-terminal transit peptide, facilitating the transportation of synthases to specific organelles, followed by the subsequent removal of transit peptides by proteases to obtain mature enzymes [29]. However, the absence of transit peptide clearance mechanisms in E. coli may result in compromised protein folding and reduced enzyme activity, thereby truncating the transit peptide, potentially leading to significant alterations in terpene titers. Monoterpene synthases were reported to be between 600 and 650 amino acids in length, and most of them had N-terminal delivery peptides required for plastid localization [30]. Increasing the titers of monoterpenes in microorganisms such as *E. coli* or *S. cerevisiae* by truncating the transit peptides of monoterpene synthases has proven to be an effective strategy. Recently, Jiang et al. made attempts to enhance geraniol production by truncating CrGES (geraniol synthase enzymes from *Catharanthus roseus*), resulting in the yield being observed at 4.45 times that of untruncated CrGES [31]. Zhou et al. modified the inactive end of the pinene synthase by tailoring the appropriate length, and the yield of $\alpha$-pinene was increased by 2.5 times [10]. Furthermore, Suzuki et al. successfully heterologously expressed $\gamma$-terpinene synthase from citrus unshiu in *E. coli* [32]. It was shown that the solubility of the truncated $\gamma$-terpinene synthase was increased.

In this study, we initially established a biosynthetic pathway for tricyclene in *E. coli* by introducing a heterologous MVA pathway, GPPS, and TS. The induction temperature and rate-limiting step for tricyclene production was determined. The tricyclene titer was greatly

improved by truncating the N-terminal of TS XP_009791411 (1411). SDS-PAGE analysis revealed that the truncation of the N-terminal of TS 1411 resulted in a significant increase in protein solubility, thus leading to an improvement in tricyclene titers.

## 2. Materials and Methods

### 2.1. Chemicals and Media

DNA polymerase was purchased from Takara (Beijing, China). T4 DNA ligase was purchased from New England Biolabs (NEB). Restriction enzymes and T4 polynucleotide kinase were purchased from Thermo Fisher Scientific (Waltham, MA, USA). The ClonExpress® II One-Step Cloning Kit for Gibson assembly was purchased from Vazyme (Nanjing, China). Plasmid extraction and gel extraction kits were provided by TIANGEN (Beijing, China). Primers and genes were synthesized by BGI (Beijing, China). A tricyclene standard was purchased from CATO (Cato Research Chemicals Inc., Canton, China), and other reagents were analytical reagents.

### 2.2. Plasmids Construction and Bacterial Strains

The strains and plasmids used in this study are listed in Table S1, and all primers used are listed in Table S2. The TSs studied here were optimized using online software (http://www.jcat.de, accessed on 20 March 2024) for codon adaption in *E. coli* strains. The plasmids' construction was performed using the *E. coli* DH5$\alpha$ strain. The expression of TSs was determined and the biosynthesis of tricyclene was performed by using *E. coli* strain BL21(DE3) and strain BS1101 [33,34], respectively. The plasmids pET21 and 3933-agGPPS were employed as the expression and tricyclene synthesis backbone vector, respectively, through ligation to TS-coding genes using the ClonExpress II One-Step Cloning Kit (Vazyme, Nanjing, China) following the manufacturer's protocols. The TS-coding genes were truncated through PCR and ligation by T4 DNA ligase. The products were digested with a DpnI restriction enzyme prior to transformation into DH5$\alpha$ chemical competent cells.

### 2.3. Shake-Flask Fermentation

The *E. coli* strain BS1101, harboring pAC-6409mva and 3933-agGPPS-tri1411, was used for the synthesis of tricyclene. For shake-flask experiments, all engineered strains were pre-cultured overnight at 37 °C and 220 rpm in LB medium and then inoculated with 10 mL of PPB (per liter: 9.8 g of $K_2HPO_4$, 5 g of beef extract, 0.3 g of ammonium ferric citrate, 2.1 g of citric acid monohydrate, 0.06 g of $MgSO_4$, 10 g of tryptone) at a ratio of 1% in 50 mL shake flasks, with 1% glucose and appropriate trace element solution [25] added to the medium. If necessary, 100 mg/L ampicillin and 34 mg/L chloramphenicol were added to the medium. When the $OD_{600}$ reached around 0.8, β-D-1-thiogalactopyranoside (IPTG) was added to the medium at a final concentration of 0.5 mM to induce gene expression, and then, a 10% (*v/v*) n-dodecane cover layer was added. After cultivating for 12 h, the culture temperature was adjusted to 30 °C with the agitation speed set at 220 rpm. All fermentation experiments were conducted in triplicate.

### 2.4. Identification and Quantification Analysis of Products

After 48 h of cultivation, the upper layer of n-dodecane was collected following centrifugation at 12,000 rpm for 1 min. Subsequently, the collected n-dodecane (100 μL) was diluted with ethyl acetate (900 μL) containing an internal standard of caryophyllene oxide at a concentration of 5 mg/L. The resulting sample was then filtered using a nylon membrane with a pore size of 0.22 μm. The samples were analyzed using GC-MS (7890A gas chromatograph and 5975C mass spectrometer, Agilent Technologies Inc., Palo Alto, CA, USA) with the following settings. A TG-5SILMS column (30 m × 250 μm i.d. × 0.25 μm film thickness) was used. The injection port temperature was 280 °C, and flow rate was 20 mL/min. The initial temperature was 50 °C, held for 0.5 min, then raised to 150 °C at a rate of 25 °C/min and held for 1 min, and then raised to 260 °C at a rate of

40 °C/min and held for 2 min. Quantification was performed using the tricyclene standard (>99%, CATO) as a control sample. The titer of tricyclene was determined by comparing the peak area ratio of tricyclene to the internal standard caryophyllene oxide, employing a standard curve of gradient tricyclene titers against resulting peak area ratios, as for other by-products.

### 2.5. Protein Analysis of Cell Lysate by SDS-PAGE

The *E. coli* strain BL21(DE3), harboring pET21-tri1411 and pET21-tri1411(44), was used for protein expression. Three single colonies were picked and transferred to 2 mL of LB medium and grown overnight at 37 °C and 220 rpm. The seed culture was inoculated to 100 mL of LB medium at a ratio of 1% (*v/v*) and continued to be cultured at 37 °C. To induce expression, 1 M IPTG was added to a final concentration of 0.3 mM when the $OD_{600}$ reached 0.6, and then, it was incubated overnight at 30 °C. Cells were collected by centrifugation at 4 °C and 4000 rpm for 10 min and resuspended in 30 mL of phosphate-buffered saline (PBS, pH 7.4). Cells were disrupted by ultrasonication on ice for 30 min. Then, 25 μL of cell lysate was taken as the total protein samples. The cell lysate was centrifugated at 8000 rpm for 30 min at 4 °C, and the supernatant was determined as the soluble protein fraction. The pellet was then resuspended in 200 μL of PBS, and the resuspension was used as the insoluble protein fraction. The Bradford protein concentration assay kit was utilized for determining the protein concentration in both the supernatant and precipitate, achieving appropriate dilution to ensure equal amounts of protein samples were applied to each lane. All diluted samples were added with 5 μL of 6× SDS-PAGE protein-loading buffer, heated at 95 °C for 5 min, and prepared for SDS-PAGE analysis. The protein concentrations were evaluated by observing the intensity of the bands.

### 3. Results

### 3.1. Characterization of Tricyclene by GC-MS

The biosynthetic pathway of tricyclene is presented (Figure 1A). To produce tricyclene in *E. coli*, we used a two-plasmid system to modulate the MVA pathway and the biosynthesis of tricyclene. The plasmid pAC-6409mva consisting of seven heterologous MVA pathway genes were shown to provide IPP and DMAPP. To convert IPP/DMAPP to tricyclene, a GPPS from *Abies grandis* and a TS were assembled into a single operon driven by the IPTG-induced trc promoter in plasmid 3933-agGPPS. The predicted TSs from different plant species are shown in Table S3. To avoid low protein expression in *E. coli*, genes coding TSs were codon adapted. After the fermentation, the n-dodecane covering agent was collected and measured by GC-MS. As a result, only TS 1411 from *Nicotiana sylvestris* presented a detectable peak at 3.38 min (Figure 1B), which was the same as the retention time of the tricyclene standard (Figure 1C). The compound was identified as tricyclene by comparing the mass spectrum of the main peak in the sample at 3.38 min with a tricyclene standard (Figure 1D,E). The relative ion abundances of molecular ions (136.1 *m/z*) and other abundant ions in the experimental samples were consistent with those of the tricyclene standard. These results demonstrate that the two-plasmid system enabled tricyclene production in E. coli, albeit at a low titer of 0.060 mg/L. In our previous experiments, we successfully synthesized α,β-Pinene and achieved a higher titer of 104.6 mg/L. The disparity in the synthesis pathway between pinene and tricyclene lies in the variation in terpene synthase, suggesting that the expression and culture conditions of TS might account for the comparatively lower titer of tricyclene [33].

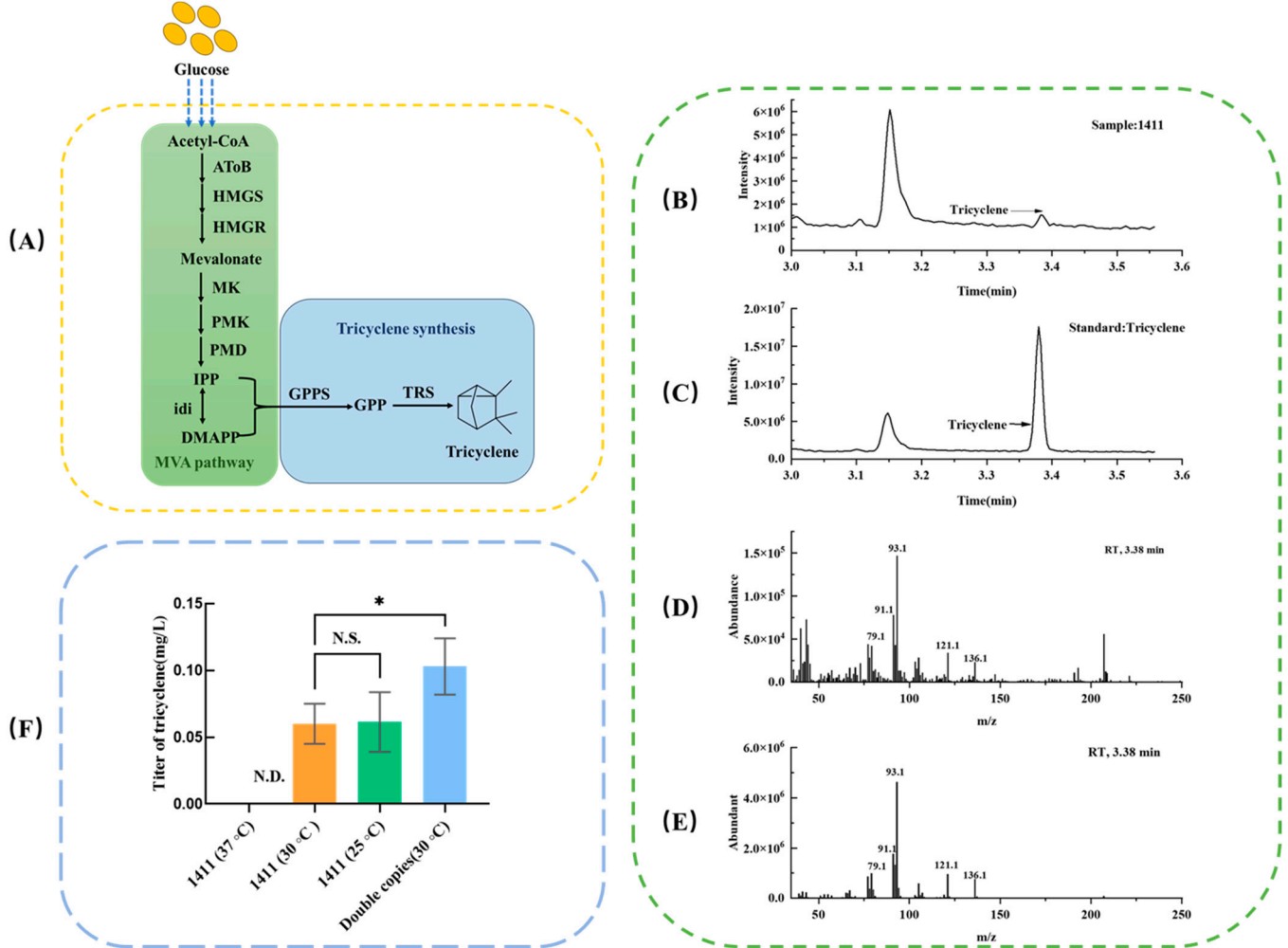

**Figure 1.** GC-MS analysis of tricyclene. (**A**) *E. coli* provides acetyl-CoA for the production of IPP and DMAPP by the MVA pathway (green box). Tricyclene was synthesized from IPP and DMAPP through a two-step conversion using GPPS and TS (blue box). (**B,C**) Total ion chromatogram (TIC) of the ZMJ01 strain, harboring one copy of the 1411-coding gene (**B**), and the tricyclene standard (**C**). (**D,E**) Mass spectrum of the tricyclene peak of the ZMJ01 strain (**D**) and the tricyclene standard (**E**). The biosynthesis of tricyclene was determined by comparing the retention time and mass spectrum with tricyclene standard. (**F**) Effects of different induction temperatures and 1411 copy number on tricyclene titers. The expression of tricyclene-producing synthases was induced by IPTG at different temperatures of 25 °C, 30 °C, and 37 °C for 48 h. Corresponding tricyclene titers are shown. Double copies of 1411-coding gene at the induction temperature of 30 °C resulted in a significantly higher tricyclene titer. N.D., not detected. The error bars represent one standard deviation for three replicates. Significance (*p*-value) was evaluated using a two-tailed t-test (* $p < 0.05$). N.S., statistically no significance.

## 3.2. Effect of Induction Temperature on Tricyclene Production

The formation of inclusion bodies due to protein misfolding may lead to inefficient catalysis and is often determined by the induction temperature [35,36]. Thus, different temperatures of 25 °C, 30 °C, and 37 °C were set to compare tricyclene titers (Figure 1F). As a result, no detectable tricyclene was produced at 37 °C, while tricyclene titers of 0.06 mg/L were achieved at 25 °C and 30 °C, indicating that a lowered temperature could enhance tricyclene titers. Given that the optimum growth temperature of the *E. coli* strain BS1101 derived from strain MG1655 was 30 °C, 30 °C was selected as the induction temperature.

### 3.3. Truncation of TS 1411

In order to elucidate the rate-limiting steps of the pathway, we introduced an additional copy of the 1411-coding gene, resulting in strain ZMJ02. This modification led to a significant improvement in titer from 0.060 mg/L to 0.103 mg/L, corresponding to a 0.67% enhancement (Figure 1F). Furthermore, it has been reported that the heterologous pathway could increase the titer of monoterpenes such as α-pinene and myrcene to 104.6 mg/L [33] and 510.38 mg/L [37], respectively, suggesting that TS was probably the rate-limiting step for the biosynthesis of tricyclene. Therefore, further improvements in the titer were conducted through reconstructions of the TSs.

Plant synthases often possess a transit peptide in the N-terminal region, which facilitates protein translocation to specific plastid compartments prior to proteolytic cleavage. Transit peptides typically consist of 30 to 80 amino acids and are characterized by an abundance of basic, slightly hydrophobic, and hydroxylated amino acids. Due to the absence of mechanisms in *E. coli* for eliminating plant transit peptides, there is a potential risk of protein misfolding. Therefore, the truncation of the transit peptide is frequently employed as a strategy to enhance synthase activity [38].

The protein structure of 1411 was predicted using Uni-Fold (Figure 2A) [39]. There was a random coil at the N-terminal of 1411 from L1 to W73, probably causing protein misfolding in *E. coli*. To determine the appropriate truncation site, the transit peptide of 1411 was predicted using the online tool TargetP 2.0 (https://services.healthtech.dtu.dk/services/TargetP-2.0/, accessed on 20 March 2024.). As a result, 1411 was predicted to have an N-terminal thylakoid luminal transfer peptide, with a cleavage site at H37. Therefore, 1411 was truncated at five different positions of P19, P24, H37, I49, and W73 at the N-terminal, obtaining 1411(19), 1411(24), 1411(37), 1411(49), and 1411(73), respectively (Figure 2B). As shown in Figure 2C, the strain containing 1411(73) was not able to produce detectable tricyclene, while the remaining four truncated 1411 synthases resulted in increased tricyclene titers at different levels, compared with wild-type 1411. The highest tricyclene titer of 38.331 mg/L was obtained for the strain containing 1411(49). Furthermore, because H37 was the predicted cutting site and the truncation at I49 led to the highest titer, 13 positions from H37 to I49 were truncated by one amino acid. The corresponding titers are shown in Figure 2D. The highest titer of tricyclene was obtained for the strain containing 1411(44), which was 47.671 mg/L, an approximately 794.5-fold improvement compared with wild-type 1411. The strains containing 1141(45), 1141(48), and 1141(49) also exhibited significantly elevated tricyclene titers of 40.708 mg/L, 38.907 mg/L, and 38.331 mg/L, respectively. This result indicated that the appropriate truncation of TS 1411 helped improve the tricyclene titer.

### 3.4. SDS-PAGE Analysis of TS 1411

To further analyze the effects of temperature and the transit peptide on protein solubility, SDS-PAGE analysis was performed. As shown in Figure S1, proteins 1411 (lane 2) and 1411(44) (lane 3), which had molecular weights of 71.5 kDa and 66.9 kDa, respectively, were successfully expressed, compared with uninduced groups (lane 1). When cultured at 37 °C (Figure S1A), 1411 was undetectable in line 4 and abundant in line 5, and 1411(44) was slightly detectable in line 6 and abundant in line 7, indicating their low solubility at 37 °C. When cultured at 30 °C (Figure S1B), 1411 was slightly detectable in lane 4, while it was abundant in lane 5, indicating that the solubility of 1411 was increased in a low level at 30 °C. By comparison, 1411(44) was abundant in both lane 6 and 7, indicating that the solubility of 1411(44) was greatly improved at 30 °C. This result was consistent with the results from shake-flask fermentation that lowering the induction temperature from 37 °C to 30 °C was able to improve the soluble expressions of 1411 and 1411(44) and further enhance tricyclene titers. Additionally, the appropriate truncation at the N-terminal of 1411 resulted in a great improvement in soluble expression level, thus leading to a 794.5-fold enhancement of the tricyclene titer.

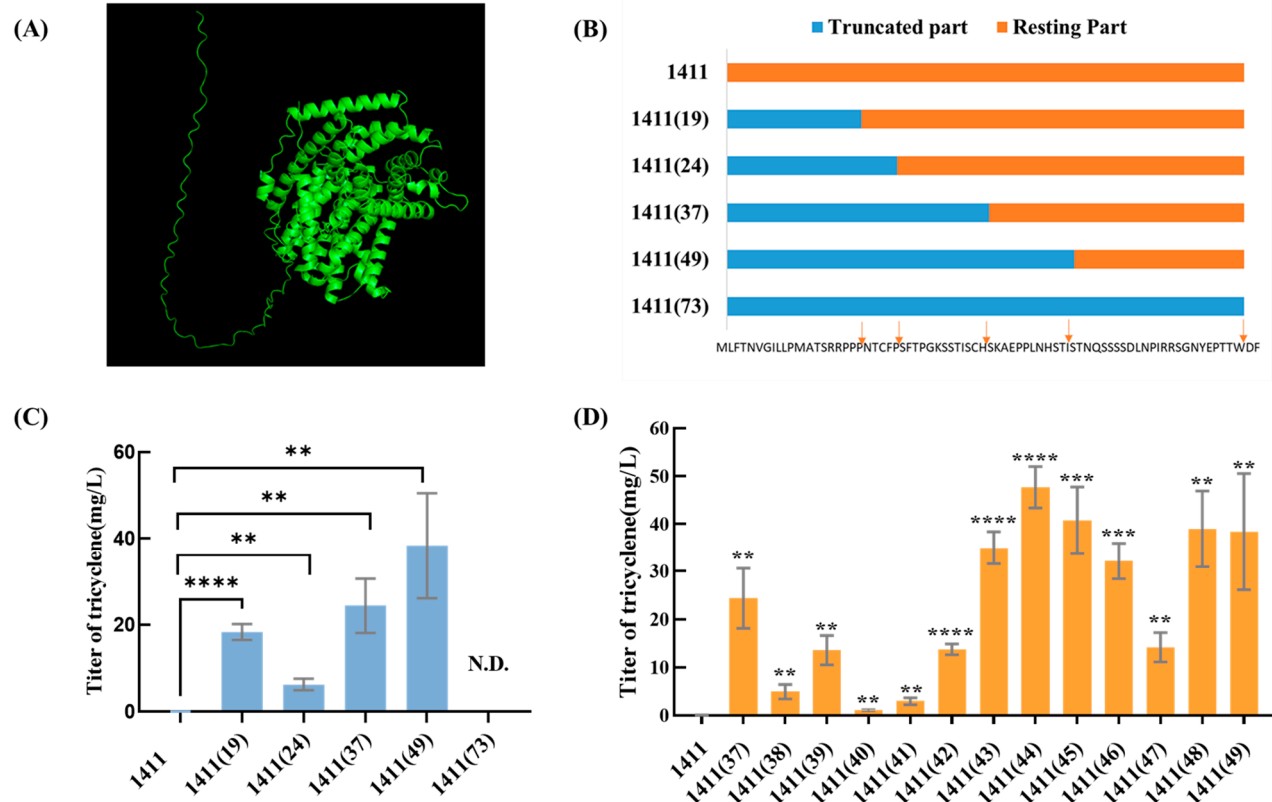

**Figure 2.** Tricyclene titers of truncated 1411 at different sites. (**A**) Protein structure prediction of 1411 using Uni-Fold. (**B**) Schematic of the N-terminal truncation of 1411. Blue bar, truncated part. Orange bar, resting part. The arrows indicated corresponding cutting sites. (**C**) Corresponding tricyclene titers at the truncation sites of P19, P24, H37, I49, and W73. (**D**) Tricyclene titers of the 13 truncation sites from H37 to I49. The significances of wild-type 1411 compared to other datasets were tested using a two-tailed *t*-test. The error bars represented one standard deviation for three replicates. Significance (*p*-value) was evaluated using a two-tailed *t*-test (** $p < 0.01$; *** $p < 0.001$, **** $p < 0.0001$).

### 3.5. Truncations of the Other TSs

As demonstrated in the GC-MS analysis (Figure 3A), *E. coli* strain BS1101 expressing TS 1411(44) exhibited the production of various monoterpenes, including tricyclene, α-pinene, β-pinene, sabinene, camphene, and limonene. To assess byproducts and titers of the remaining 15 TSs investigated in this study, truncations of the transit peptide were performed. Due to the high sequence similarities of these TSs and 1411, all of them were truncated by 44 amino acids and subsequently subjected to shake-flask fermentation under identical conditions. The results of the shake-flask fermentation are shown in Figure 3B, including titers of tricyclene and the other monoterpene byproducts. The strains containing 4091(44), 4926(44), 5066(44), 8653(44), 9309(44), 5307(44), and 8401(44) were capable of synthesizing detectable tricyclene, with 8653(44) producing 4.329 mg/L of tricyclene, which was the second highest titer after 1411(44). These TSs were not able to produce tricyclene before being truncated, indicating that the truncation of the transit peptide had a significant enhancement in the synthesis of tricyclene. The rest of the TSs in Table S3 could not produce detectable tricyclene after being truncated. In addition, these truncated TSs also produced other monoterpene by-products including α-pinene, β-pinene, camphene, sabinene, and limonene in *E. coli* strain BS1101. The proportions of tricyclene among all monoterpenes varied across different TSs, as presented in Table S4. Among these, 1411(44) exhibited the highest proportion of tricyclene at 60.8%. The proportion of by-products derived from different sources of TPSs varied significantly. For instance, β-pinene and limonene were the predominant by-products of 1411(44) and 8653(44), whereas camphene, β-pinene, and

limonene were the main by-products of 5307(44) and 8401(44). In conclusion, these findings suggest that truncation at the N-terminal of TSs effectively enhances tricyclene production in *E. coli*.

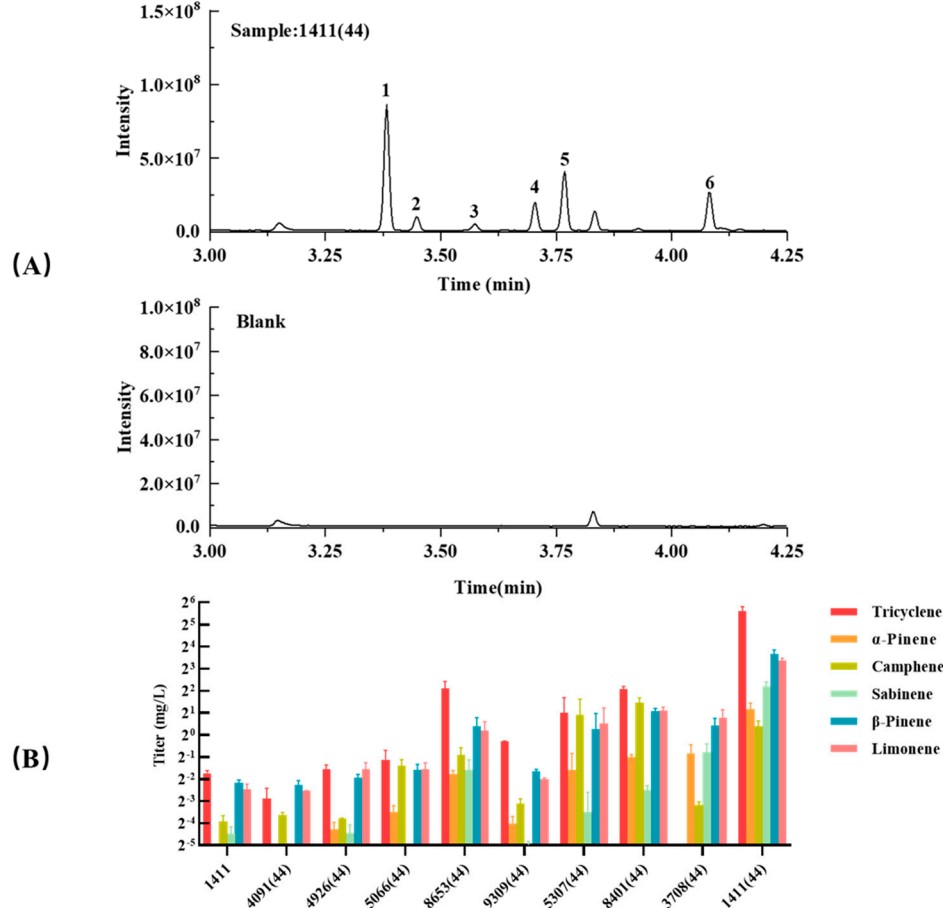

**Figure 3.** Product analysis of the truncated TSs. (**A**) GC-MS analysis of 1411(44) products. Peak 1, tricyclene; Peak 2, α-pinene; Peak 3, camphene; Peak 4, sabinene; Peak 5, β-pinene; Peak 6, limonene. Blank represents a mixture of n-dodecane and ethyl acetate. (**B**) Product titers of the truncated TSs. The truncated TSs producing detectable products in this study. The error bars represent one standard deviation for three replicates.

## 4. Discussion

Tricyclene, a tricyclic monoterpene, has broad application prospects in many fields, including biofuel and medicine. However, due to the scarcity of tricyclene, its use needs to be further explored. In this study, we reported the construction of a heterologous biosynthetic pathway of tricyclene in *E. coli* by introducing TSs from different plant species. It was determined by increasing the copy number of the TS-coding gene that the TS was the rate-limiting step for tricyclene biosynthesis. Low temperatures could reduce the inclusion bodies of *E. coli*, thereby increasing production [25]. The optimal culture temperature for tricyclene synthesis was determined to be 30 °C, as further reductions in incubation temperature did not yield any improvement in tricyclene titer.

The low activity of tricyclene synthase leads to its low biosynthetic potency, which needs further optimization to meet the requirements of large-scale industrial production. The tricyclene titer was increased from 0.060 mg/L to 47.671 mg/L, a 794.5-fold improvement, through truncating 44 amino acids of the N-terminal random coil of TS 1411. The titers of byproducts including α-pinene, β-pinene, sabinene, camphene, and limonene were also increased. Removing transit peptides could improve protein expression in the *E. coli* system without affecting its enzyme activity [40]. Suzuki et al. introduced γ-terpinene

synthase in *E. coli* to synthesize γ-terpinene; the results of SDS-PAGE showed that the formation of inclusion bodies could be reduced by removing the transit peptide [32]. In this study, SDS-PAGE analysis revealed that by lowering the induction temperature to 30 °C and truncating the transit peptide, the soluble expression level of TS was effectively improved, indicating that the enhancement of the tricyclene titer was probably caused by improved TS solubility. Our findings indicate that the appropriate truncation of plant synthase transit peptides can improve their soluble expression and product titers in *E. coli*, which lacks a mechanism for clearing plant transit peptides.

According to the results above, the truncated region referred to the transit peptide segment, which impacted the soluble expression of TS. Truncating the transit peptide significantly enhanced the solubility of TS and subsequently improved tricyclene's synthetic titer. In cases where truncation was insufficient, the incomplete removal of the transit peptide still affected protein solubility, resulting in a change in tricyclene's synthetic titer due to variations in TS solubility during gradient truncation. If the truncated region was excessively long (e.g., 1411(73)), it potentially disrupted a domain within TS responsible for catalyzing tricyclene synthesis, thereby reducing its synthetic titer instead. Consequently, we hypothesized that achieving the highest tricyclene titer required optimal TS solubility without compromising the integrity of the domain involved in tricyclene synthesis.

To synthesize tricyclene, we retrieved 16 synthases predicted to synthesize tricyclene from public databases (Table S3) and introduced them into *E. coli*. The fermentation results showed that only 1411 could synthesize detectable tricyclene. Due to the high similarity of the protein sequences and predicted conformations of the retrieved 16 TSs, the transit peptides of the other 15 TSs should be consistent with 1411. Therefore, the other 15 truncated TSs were subjected to shake-flask fermentation after being truncated by 44 amino acids. However, despite their high similarities with TS 1411, these 15 TSs exhibited significantly lower titers of tricyclene production. Among all monoterpene products in different TSs, 1411(44) exhibited the highest proportion of tricyclene, accounting for 60.8% (Table S4). Furthermore, the truncated forms of these 16 TSs displayed distinct proportions of by-products. While β-pinene and limonene were the main by-products produced by 8653 and 1411, respectively, camphene was predominantly generated by 5307 and 8401. Interestingly, it was found that 3708 did not function as a tricyclene synthase but only yielded various by-products. Given the high similarities of these TSs, the titers of tricyclene and by-products were quite different, suggesting that the divergent amino acids of 1411 probably play an important role in improving the production and purity of tricyclene. This finding may be useful for the directed evolution and other engineering strategies of TS in the future studies.

## 5. Conclusions

This study reported the introduction of TS 1411 to construct a two-plasmid system to modulate the MVA pathway and achieve tricyclene biosynthesis in *E. coli*. By optimizing the culture temperature, increasing the copy number of the TS-coding gene, and truncating the random coil sequences at the N-terminal of TSs, we achieved a remarkable 794.5-fold improvement in the tricyclene titer compared to its wild type, with an increase from 0.060 mg/L to 47.671 mg/L. The results of SDS-PAGE analysis revealed that the solubility of TS was significantly enhanced by reducing the induction temperature and truncating the TSs, indicating a close relationship between tricyclene titer and TS solubility. Furthermore, varying degrees of increases in tricyclene titers were observed upon truncation of other TSs. To the best of our knowledge, this report presents the highest recorded titer of tricyclene and specificity of tricyclene synthase to date.

**Supplementary Materials:** The following supporting information can be downloaded at: https://www.mdpi.com/article/10.3390/fermentation10030173/s1, Table S1: Strains and plasmids used in this study; Table S2: Primers used in this study. Table S3: Tricyclene synthases derived from different species. Table S4: Tricyclene and by-product titers of different TSs. Figure S1: SDS-PAGE analysis of wild-type 1411 and 1411(44).

**Author Contributions:** Conceptualization, M.Z., S.B., H.J. and W.Z.; methodology, M.Z., S.B., X.Z. and W.Z.; software, S.B.; validation, M.Z., S.B., J.L., C.C. and P.H.; formal analysis, M.Z., S.B., F.W. and X.W.; investigation, M.Z. and S.B.; resources, G.Y.; data curation, M.Z., S.B., J.L., F.W., G.Y. and X.W.; writing—original draft preparation, M.Z.; writing—review and editing, M.Z., S.B., H.J., X.Z. and W.Z.; visualization, M.Z., S.B., J.L. and P.H.; supervision, H.J., X.Z. and W.Z.; project administration, W.Z.; funding acquisition, H.J., X.Z. and W.Z. All authors have read and agreed to the published version of the manuscript.

**Funding:** This research was funded by National Natural Science Foundation of China Grant 32300066.

**Institutional Review Board Statement:** Not applicable.

**Informed Consent Statement:** Not applicable.

**Data Availability Statement:** Data is contained within the article and Supplementary Materials.

**Conflicts of Interest:** The authors declare no conflicts of interest.

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
