# Peer review of "The Biosynthesis of the Monoterpene Tricyclene in E. coli through the Appropriate Truncation of Plant Transit Peptides"

_fermentation, doi:10.3390/fermentation10030173_

Round 1

Reviewer 1 Report

Comments and Suggestions for Authors

This manuscript by Zhao et al. tried to deal with the production of tricyclene in E. coli. The authors used the properly design experiments to meet the objective of the study. However, they are strongly suggested to make proper discussion and make data analysis more solid. Furthermore, there are many unbalanced sentences, grammatical errors and typos giving a bumpy flow to the reading. The authors are strongly encouraged to rewrite the whole manuscript for a cohesive and concise presentation of this interesting work. Thus, the reviewer thought this paper need thorough revision before considering for the publication in Fermentation.

Major:                                                  

Comment 1: In section 3.1, authors showed the possibility of tricyclene production in E. coli through overexpression of MVA pathway, GPPS and TS. However, the obtained titer was very low. Authors are suggested to discuss the possible reason for this low production and how it can be improved in addition to GC-MS raw data analysis.

Comment 2: In section 3.2, authors stated that insoluble expression of pathway protein could be one reason for the low titer of tricyclene. However, no supporting data was provided. Authors are suggested to provide the SDS-PAGE analysis of pathway protein to support their hypothesis. In addition, authors are suggested to show the result and discuss the improvement made in soluble expression of protein by reducing the temperature.

Comment 3: In section 3.3, authors showed the improvement in production of tricyclene by overexpression of truncated TS. The production of tricyclene varied depending upon the truncation sites. However, author did not discuss how truncation of TS affect the tricyclene. Authors are strongly suggested to discuss the possible reasons.

Comment 4: Authors are suggested to move SDS-PAGE analysis of TS 1411 to supplementary.

Comment 5: In section 3.5, authors stated they studied the 15 TS. The production profile of monoterpene widely varied depending upon the TS. But the data analysis was very poor. Authors are strongly suggested to present and discuss their data more clearly.

Comments on the Quality of English Language

Extensive editing of English language required

Author Response

Dear reviewer:

Thank you very much for taking the time to review this manuscript entitled “Biosynthesis of the monoterpene tricyclene in E. coli through appropriate truncation of plant transit peptides” (fermentation-2893127). Those comments are all valuable and very helpful for revising and improving our paper. We have studied comments carefully and have made correction which we hope meet with approval. Revised portion are marked in red in the paper. The main corrections in the paper and the responds to the reviewer’s comments are as following:

Comment 1: In section 3.1, authors showed the possibility of tricyclene production in E. coli through overexpression of MVA pathway, GPPS and TS. However, the obtained titer was very low. Authors are suggested to discuss the possible reason for this low production and how it can be improved in addition to GC-MS raw data analysis.

Response: Thank you for your reminder. In section 3.1 of the manuscript, we have added a discussion on the possible reasons for the low titer of tricyclene. The added content is in L 188-192. The specific reasons are detailed in sections 3.3 and 3.4 of the manuscript.

Comment 2: In section 3.2, authors stated that insoluble expression of pathway protein could be one reason for the low titer of tricyclene. However, no supporting data was provided. Authors are suggested to provide the SDS-PAGE analysis of pathway protein to support their hypothesis. In addition, authors are suggested to show the result and discuss the improvement made in soluble expression of protein by reducing the temperature.

Response: Thank you for your reminder. SDS-PAGE analysis of TS and related discussion have been provided in section 3.4. We believe that it is necessary to present and discuss this problem in a specific section.

Comment 3: In section 3.3, authors showed the improvement in production of tricyclene by overexpression of truncated TS. The production of tricyclene varied depending upon the truncation sites. However, author did not discuss how truncation of TS affect the tricyclene. Authors are strongly suggested to discuss the possible reasons.

Response: The truncated region refers to the transit peptide segment, which impacts the soluble expression of TS. Truncating the transit peptide significantly enhances the solubility of TS and subsequently improves tricyclene's synthetic titer. In cases where truncation is insufficient, incomplete removal of the transit peptide still affects protein solubility, resulting in a change in tricyclene's synthetic titer due to variations in TS solubility during gradient truncation. If the truncated region is excessively long (e.g., 1411(73)), it may potentially disrupt a domain within TS responsible for catalyzing tricyclene synthesis, thereby reducing its synthetic titer instead. Consequently, we hypothesize that achieving the highest tricyclene titer requires optimal TS solubility without compromising the integrity of the domain involved in tricyclene synthesis. A corresponding discussion has been included at L332-341.

Comment 4: Authors are suggested to move SDS-PAGE analysis of TS 1411 to supplementary.

Response: Thank you for your suggestion. The resulting images have been included in the supporting information.

Comment 5: In section 3.5, authors stated they studied the 15 TS. The production profile of monoterpene widely varied depending upon the TS. But the data analysis was very poor. Authors are strongly suggested to present and discuss their data more clearly.

Response: Thank you for your reminder. We have optimized Figure 3B and added a new table S4 in the supplementary to further analyze the data in section 3.5.

We apologize for the poor language of our manuscript. We worked on the manuscript for a long time and the repeated addition and removal of sentences and sections obviously led to poor readability. We really hope that the flow and language level have been substantially improved.

In all, we found the reviewer’s comments quite helpful and revised the manuscript point-by-point. Thank you again for your help!

Reviewer 2 Report

Comments and Suggestions for Authors

In the manuscript you suggest that this is the first report of recombinant production of tricyclene. This is not strictly true as Codexis have a patent which shows production of tricyclene, but it was only a minor component of the products they observed. They seemed to be particularly interested in the fuel properties of tricyclene so this should be referenced. The heterologous production of a "specific" tricyclene synthase appears to be novel. I say "specific" because they all produce other products.

Evidence of problems due the expression of the transit peptide is well established, so there is a question as to why you did this. There is no advantage to heterologous production, so most people start with expression of a truncated peptide. The more interesting fine-truncation findings are still valid.

You discussion is largely a summary of results. Can you look at the predicted structure of 141137) and work out why 1411(44),(45). (48) and (49) should fold better. Also can you look at the mechanisms by which tricyclene is predicted to form to work out why the other by-products are appearing. Do you think this is a natural phenomenon or the result of heterologous expression?

Specific points:

L64: Vincent et al should be Martin et al

L102-103: Give full names (and abbreviation) and places for TIANGEN, BGE and CATO.

Table S3: The order of presentation appears random. I suggest that you group by genus (eg all Nicotiana together)

L197-198: Presentation of data to 3dp is not justified given the size of the error bars. These numbers are identical

L223: You have omitted 1411(37) !

Fig 2 and text: The error bars of 1411(41) overlap with 1141(45), 1141(48) and 1141(49). Therefore, they cannot be differentiated

Fig 3 (and methods): You do not appear to have loaded the same amount of protein in each lane, so comparison of abundance in each lane is not simple.

Comments on the Quality of English Language

The manuscript is readable but there are a number of places where there is incorrect usage or grammar (eg line 18, "appropriate" should be "optimal", line 41, the use of fossil fuels has not led to "energy crisis" - but it is leading to a "climate crisis")

Author Response

Dear reviewer:

Thank you for reviewing my manuscript “Biosynthesis of the monoterpene tricyclene in E. coli through appropriate truncation of plant transit peptides” (fermentation-2893127) and providing valuable suggestions and important guiding significance to our researches. We have carefully considered your proposal and made necessary modifications, highlighted in red. Below are our responses to the reviewers' feedback:

Comment 1: In the manuscript you suggest that this is the first report of recombinant production of tricyclene. This is not strictly true as Codexis have a patent which shows production of tricyclene, but it was only a minor component of the products they observed. They seemed to be particularly interested in the fuel properties of tricyclene so this should be referenced. The heterologous production of a "specific" tricyclene synthase appears to be novel. I say "specific" because they all produce other products.

Response: Thank you for your kind reminder. We have carefully read the patent you mentioned, which is a valuable reference, so we have revised the description of “the first synthesis of tricyclene in E. coli” in the manuscript as “the highest titer of heterologous synthesis of tricyclene in E. coli”, and revise it at L20 and L370-371. We also cited this patent in the manuscript at L62-66. We added the proportion of tricyclene in all products in the supplementary table. Tricyclene accounted for 60.1 % of the 1411 (44) products we screened, which was much higher than the highest value in patents. Therefore, this paper found the synthases with the highest titer and specificity of tricyclene synthesis so far. Relevant analysis and conclusions are added in the L349-351.

Comment 2: Evidence of problems due the expression of the transit peptide is well established, so there is a question as to why you did this. There is no advantage to heterologous production, so most people start with expression of a truncated peptide. The more interesting fine-truncation findings are still valid.

Response: As you mentioned, the expression of the transit peptide is not uncommon in the study of heterologous expression, we initially conducted a preliminary truncation, and the increase in the yield of tricyclene was astonishing, which is why we are considering further precise truncation.

Comment 3: You discussion is largely a summary of results. Can you look at the predicted structure of 1411(37) and work out why 1411(44),(45). (48) and (49) should fold better. Also can you look at the mechanisms by which tricyclene is predicted to form to work out why the other by-products are appearing. Do you think this is a natural phenomenon or the result of heterologous expression?

Response: It is well known that terpene synthases naturally convert substrates into various related products, a process determined by their specificity. It has been reported that the occurrence of by-products in heterologous expression of monoterpene synthases is a common phenomenon[1,2]. However, despite previous reports, no TS with high specificity for tricyclene synthesis has been identified. Therefore, our research on by-products aimed to highlight the specificity of the screened tricyclene synthases and identify one that is more specific for tricyclene. As for the mechanism of TS synthesis by-products in E. coli, we believe that in vitro enzymatic experiments of TS are needed to verify whether the synthesis by-products of TS are natural phenomena. But no relevant experimental studies have yet been carried out in this paper, so we cannot give the corresponding conclusions. In our future research, we will carry out relevant experimental verification.

Specific points:

Comment 4:L64: Vincent et al should be Martin et al

Response: We are very sorry for our incorrect writing and it is rectified at L71.

Comment 5:L102-103: Give full names (and abbreviation) and places for TIANGEN, BGE and CATO.

Response: We are very sorry for our incorrect writing and it is rectified at L109-111.

Comment 6:Table S3: The order of presentation appears random. I suggest that you group by genus (eg all Nicotiana together)

Response: As you mentioned, the random sorting of Table S3 is not justified, hence we have revised Table S3 and sorted it by species.

Comment 7:L197-198: Presentation of data to 3dp is not justified given the size of the error bars. These numbers are identical

Response: Thank you for your reminder. We have revised the description at L212.

Comment 8:L223: You have omitted 1411(37) !

Response: We are very sorry for our incorrect writing and it is rectified at L239.

Comment 9:Fig 2 and text: The error bars of 1411(41) overlap with 1141(45), 1141(48) and 1141(49). Therefore, they cannot be differentiated

Response: As you mentioned, the error bars of 1411 (44) and those of 1141 (45), 1141 (48), and 1141 (49) are overlapped, which means their titers are very close. But in Fig2, we compared the significance of the wild-type 1411 with other datasets. The TS 1411 (44) was selected for subsequent experiments solely based on its highest titer mean. In addition, we added a description of the yield of 1141 (45), 1141 (48), and 1141 (49) in L248-249 of the manuscript.

Comment 10:Fig 3 (and methods): You do not appear to have loaded the same amount of protein in each lane, so comparison of abundance in each lane is not simple.

Response: We thank the reviewer of pointing out this issue. We repeated the SDS-PAGE experiment and have loaded the same amount of protein in each lane.

We tried our best to improve the manuscript and made some changes in the manuscript. These changes will not influence the content and framework of the paper. And here we did not list the changes but marked in red in revised paper.

We appreciate for Reviewers’ warm work earnestly, and hope that the correction will meet with approval. Moreover, we have made diligent efforts to refine the language in the revised manuscript.

Once again, thank you very much for your comments and suggestions.

  1. Formighieri, C.; Melis, A. Carbon Partitioning to the Terpenoid Biosynthetic Pathway Enables Heterologous β-Phellandrene Production in Escherichia Coli Cultures. Arch Microbiol 2014, 196, 853–861, doi:10.1007/s00203-014-1024-9.
  2. Toogood, H.S.; Tait, S.; Jervis, A.; Ní Cheallaigh, A.; Humphreys, L.; Takano, E.; Gardiner, J.M.; Scrutton, N.S. Natural Product Biosynthesis in Escherichia Coli. In Methods in Enzymology; Elsevier, 2016; Vol. 575, pp. 247–270 ISBN 978-0-12-804584-8.

Round 2

Reviewer 1 Report

Comments and Suggestions for Authors

Authors responded the reviewers comments properly and modified the manuscript accordingly. The paper is acceptable for publication.

Comments on the Quality of English Language

Minor editing is needed.